# The Complete Mitochondrial Genome of Four Hylicinae (Hemiptera: Cicadellidae): Structural Features and Phylogenetic Implications

**DOI:** 10.3390/insects11120869

**Published:** 2020-12-07

**Authors:** Jiu Tang, Weijian Huang, Yalin Zhang

**Affiliations:** Key Laboratory of Plant Protection Resources and Pest Management, Ministry of Education, Entomological Museum, College of Plant Protection, Northwest A&F University, Yangling 712100, China; tangjiu@nwafu.edu.cn (J.T.); jackyhuang@nwafu.edu.cn (W.H.)

**Keywords:** mitochondrial DNA, leafhopper, Hylicinae, phylogeny

## Abstract

**Simple Summary:**

Hylicinae, containing 43 described species in 13 genera of two tribes, is one of the most morphologically unique subfamilies of Cicadellidae. Phylogenetic studies on this subfamily were mainly based on morphological characters or several gene fragments and just involved single or two taxa. No mitochondrial genome was reported in Hylicinae before. Therefore, we sequenced and analyzed four complete mtgenomes of Hylicinae (*Nacolus tuberculatus*, *Hylica paradoxa*, *Balala fujiana*, and *Kalasha nativa*) for the first time to reveal mtgenome characterizations and reconstruct phylogenetic relationships of this group. The comparative analyses showed the mtgenome characterizations of Hylicinae are similar to members of Membracoidea. In phylogenetic results, Hylicinae was recovered as a monophyletic group in Cicadellidae and formed to the sister group of Coelidiinae + Iassinae. These results provide the comprehensive framework and worthy information toward the future researches of this subfamily.

**Abstract:**

To reveal mtgenome characterizations and reconstruct phylogenetic relationships of Hylicinae, the complete mtgenomes of four hylicine species, including *Nacolus tuberculatus*, *Hylica paradoxa*, *Balala fujiana,* and *Kalasha nativa*, were sequenced and comparatively analyzed for the first time. We also carried out the richest (11) subfamily sampling of Cicadellidae to date, and reconstructed phylogenetic relationships of Membracoidea among 61 species based on three datasets using maximum likelihood and Bayesian inference analyses. All new sequenced mtgenomes are molecules ranging from 14,918 to 16,221 bp in length and are double stranded, circular in shape. The gene composition and arrangement of these mtgenomes are consistent with members of Membracoidea. Among 13 protein-coding genes, most show typical ATN start codons and TAR (TAA/TAG) or an incomplete stop codon T–, and several genes start by TTG/GTG. Results of the analysis for sliding window, nucleotide diversity, and nonsynonymous substitution/synonymous substitution indicate *cox1* is a comparatively slower-evolving gene while *atp8* is the fastest gene. In line with previous researches, phylogenetic results indicate that treehopper families are paraphyletic with respect to family Cicadellidae and also support the monophyly of all involved subfamilies including Hylicinae. Relationships among the four hylicine genera were recovered as (*Hylica* + (*Nacolus* + (*Balala* + *Kalasha*))).

## 1. Introduction

The infraorder Cicadomorpha in Hemiptera includes three superfamilies: Cicadoidea, Cercopoidea, and Membracoidea. The last one, comprising Cicadellidae, Myerslopiidae, and three treehopper families (Membracidae, Melizoderidae, and Aetalionidae), is the most diverse among these groups [1]. The leafhopper family Cicadellidae, containing >2600 genera and about 21,000 species distributed worldwide, is one of the largest families of Hemiptera [2].

One such group, the hylicine leafhoppers, is one of the most morphologically unique cicadellids and was once treated as a tribe of Cicadellinae [3] as well as an independent family with respect to Cicadellidae [4]. However, it is currently recognized as a cicadellid subfamily based on morphological and molecular evidence [1,5,6]. Among subfamilies of Cicadellidae, Hylicinae can be distinguished from the others easily by the following combination of characters: body brown to black covered with scale-like setae, head usually produced, ocelli on the crown close to eyes, forewing appendix relatively wide. This subfamily is very small and is comprised of 43 described species in 13 genera assigned to two tribes [6,7]. Most (11) genera of Hylicinae are restricted to the Oriental and eastern Palearctic regions. The two genera *Wolfella* Spinola, 1850, and *Karasekia* Melichar, 1926, are distributed in the Ethiopian region [8]. Hylicines have been found on grasses and trees in moistly forested areas and on chili in farmland [9,10]. Many leafhoppers spread viruses causing plant diseases, but that ability has not been found in this group yet [11].

Research on this group has mainly focused on species taxonomy [9,12,13,14,15,16]. Phylogenetic analyses are relatively rare due to the difficulty of obtaining specimens due to their limited of distribution, scarcity of species, etc. One phylogenetic study of Membracoidea based on *28S* rDNA included one hylicine exemplar [5]. Results showed that the position of Hylicinae varied and the relationship of Hylicinae to other subfamilies had not been resolved. Recently, a phylogenetic analysis using DNA sequences including nearly hundreds of thousands aligned nucleotides from hundreds of loci also suggested that Hylicinae, represented by two species of one tribe, is monophyletic with strong branch support. The results were also unable to resolve the relationships of Hylicinae and related subfamilies [17]. The monophyly and the phylogenetic status in Cicadellidae of Hylicinae need further study based on more exemplars and using different methods.

The typical insect mtgenome is a double-stranded circular molecule and generally contains 37 genes: 13 protein coding genes (PCGs), 22 transfer RNA genes (tRNAs), two ribosomal RNA genes (rRNAs), and one non-coding region (CR) [18,19,20]. The genome-level features that mtgenomes can offer include base composition, gene arrangement, genetic codon variation, and secondary structures of tRNA and rRNA [21,22]. More and more insect mitochondrial genomes have been obtained with the development of sequencing technology. To date, the mtgenomes of Membracoidea have been determined and contributed to GenBank (https://www.ncbi.nlm.nih.gov) more than 150. Unfortunately, there is no mtgenome or even a partial one for the subfamily Hylicinae. Simple genetic structure, small size, strict orthologous genes, high genome copy numbers, less recombination, and fast evolutionary rate are characteristics of mtgenomes [18,19,23]. Those characteristics make mtgenome a new tool. In the past decade, mtgenomes have proven to be a useful source for studies on inter- and intra-specific variation as well as the systematics and phylogeny of arthropods at different levels including leafhoppers [24,25,26,27,28,29,30]. Therefore, mtgenomes may be a better method to analyze the taxonomic and phylogenetic status of Hylicinae. Meanwhile, a lack of mtgenome data for hylicine species impedes a more complete phylogenetic analysis of Membracoidea. So it is necessary and urgent to fill in the gaps in the mtgenome of this subfamily.

In this study, we sequenced and annotated mtgenomes of four Hylicinae leafhoppers (*Nacolus tuberculatus* (Walker, 1858), *Hylica paradoxa* Stål, 1863, *Balala fujiana* Tang and Zhang, 2020 and *Kalasha nativa* Distant, 1908). Using the mtgenomes of membracoids including these four species, the phylogeny of Membracoidea was reconstructed. The aims of this study were to (1) figure out the mitochondrial structure including gene order, nucleotide composition, codon usage, tRNA secondary structure, gene overlap, and the non-coding control region of hylicine species (2) verify the monophyly of subfamily Hylicinae and explore the phylogenetic relationships of Hylicinae to other major lineages.

## 2. Materials and Methods

### 2.1. Sample Collection and DNA Extraction

*Nacolus tuberculatus* was captured in Qingquan Gully (109.4412° E, 36.2472° N), Ganquan County, Yan’an City, Shaanxi Province, China, on 19 June 2019. *Hylica paradoxa* was collected in Menglun County (101.2552° E, 21.9412° N), Yunnan Province, China, on 4 May 2009. The *B. fujiana* specimen was captured in Mengla County (101.5682° E, 21.4822° N), Yunnan Province, China, on 27 April 2019, while the *K. nativa* specimen was collected in Jianfengling (110.3491° E, 19.9770° N), Hainan Province, 975 m, China, on 15 August 2010. Four species of Hylicinae were identified by external morphology and genitalia [7,31,32,33]. All samples were kept in tubes fulling of absolute ethyl alcohol at minus 20 °C. Voucher specimens were stored in the Entomological Museum, Northwest A&F University (NWAFU), Yangling, Shaanxi, China.

Total genomic DNA was extracted using the Easy Pure Genomic DNA Kit following the manufacturer’s instructions from the thoracic muscle.

### 2.2. Mtgenome Sequences Acquisition and Bioinformatic Analyses

The whole mtgenomes of four species in Hylicinae were generated by NGS. The paired-end clean reads were assembled by employing the mtgenomes of *Evacanthus heimianus* Kuoh, 1981, (MG813486) and *N. tuberculatus* (MW218663) as references in Geneious10.0.5. A total of 16,019,948/22,247,400/24,753,222/15,786,834/paired-end clean reads were assembled with trim sequence and under medium-low sensitivity to assemble. Then we generated consensus sequence and used MAFFT (within Geneious 10.0.5., Biomatters, Auckland, New Zealand) to align the consensus sequences and reference sequences for annotation.

The annotation of four mtgenomes was accomplished in Geneious10.0.5. Twenty-two tRNAs were found by MITOS [34] and tRNAscan-SE v1.21 [35] under the invertebrate mitochondrial genetic code. Two rRNAs were predicted by aligning with a reference mtgenomes and its boundaries were defined by the adjacent tRNAs. All 13 PCGs were determined by the ORF Finder employing codon Table 5 (invertebrate mitochondrion) and comparison with the homologous sequence of other Cicadellidae species. The different types of repeat tandem units were identified using the Tandem Repeats Finder server [36]. The mtgenome map was produced using CGView [37].

The analyses of the four newly sequenced mtgenomes of Hylicinae, including the base composition and relative synonymous codon usage (RSCU), were calculated using MEGA v 7.313 [38] incorporated into PhyloSuite v 1.2.2 [39]. The AT and GC base compositional skewness were computed employing the formulas (A – T)/(A + T) and (G – C)/(G + C), respectively [40].

The Pi value (nucleotide diversity), the ratios between non-synonymous (Ka) and synonymous (Ks) substitutions rates for each PCG, and a sliding window analysis (a sliding window of 200 bp and step size of 20 bp) of 13 PCGs among four Hylicinae species were conducted by DnaSP 6.0 [41]. Mean genetic distances among species were analyzed under Kimura-2-parameter using MEGA v 7.313 [38]. The genetic distances, and Ka/Ks ratios were graphically produced by Prism 6.01. The four mtgenome sequences of Hylicinae (*N. tuberculatus*, *H. paradoxa*, *B. fujiana*, and *K. nativa*) were registered in GenBank as MW218663, MW218660, MW218661, and MW218662, respectively (Table 1).

### 2.3. Phylogenetic Analyses

Among the 150 available Membracoidea mtgenomes, there are some repeats of same species. We chose the most complete and highest quality one among these repeats. The aim of this study is to explore the monophyly of Hylicinae and the phylogenetic relationships of Hylicinae to other major subfamilies. So we should choose moderate species representing each subfamily/tribe rather than all. In phylogenetic analyses, we chose 50 Membracoidea samples (covering 44 leafhoppers, six treehoppers) as ingroup and eleven previously available samples of the other three suborders in Hemiptera as outgroups. Four mtgenomes of Hylicinae species (*H. paradoxa*, *K. nativa*, *N. tuberculatus*, and *B. fujiana*) were sequenced in this study while the other sequences were acquired from the GenBank (Table 1).

PhyloSuite v 1.2.2 was used for genes extraction [39]. All PCGs and rRNAs were aligned individually with the G-INS-I algorithm and Q-INS-I algorithm respectively in MAFFT v 7.313 online service [69]. Ambiguous sites and gaps in the alignments were removed using GBlocks v 0.91b [70].

Three optimized datasets were merged in PhyloSuite v 1.2.2: (1) PCG123 matrix, including 3 codon positions of 13 PCGs (10,725 bp); (2) PCG123RNA matrix, including 3 codon positions of the 13 PCGs and two rRNAs (11,650 bp); and (3) AA matrix, including amino acid sequences of 13 PCGs (3277 bp) [39]. Maximum likelihood (ML) and Bayesian inference (BI) analyses were used for phylogenetic reconstruction of all matrices. The best partitioning strategies were inferred in PartitionFinder 2.1.1 [71] incorporated into PhyloSuite v1.2.2 [39] with the greedy algorithm and BIC criterion (Appendix A). ML analyses were implemented in IQ-TREE v 1.6.8 [72] under an edge-linked partition model. Branch support analyses were conducted under 10,000 ultrafast bootstrap replicates (UFB) [73]. BI analyses were conducted in MrBayes 3.2.6 [74], as implemented in the CIPRES Science Gateway [75]. Each BI analysis involved 5,000,000–20,000,000 generations. The convergence of the independent runs was indicated by a standard deviation of split frequencies < 0.01 and effective sample size (ESS) > 200. A consensus tree was computed from the trees after the initial 25% trees of each MCMC run were discarded as burn-in.

## 3. Results and Discussion

### 3.1. Genome Organization and Base Composition

The mtgenomes of *N. tuberculatus* (15,737 bp), *H. paradoxa* (14,762 bp), *B. fujiana* (16,221 bp), and *K. nativa* (15,716 bp) were single, closed circular double-stranded molecules (Figure 1 and Figure 2). Among the four mtgenomes of Hylicinae, *B. fujiana* was the largest in 16,221 bp, while *H. paradoxa* had the smallest mtgenome of 14,762 bp. Length variation of Membracoidea mtgenomes appears causing by variation in the length of the control region [59,61]. Typical 37 animal mitochondrial genes (13 PCGs, 22 tRNAs, and two rRNAs) and one non-coding region were detected in all mtgenomes of Hylicinae (Figure 1 and Figure 2). The J-strand encoded 23 genes (nine PCGs and 14 tRNAs), while the remaining 14 genes (four PCGs, eight tRNAs, and two rRNAs) were transcribed on the J-strand (Appendix A). All the sequenced mtgenomes of Hylicinae possessed the same ancestral gene order as the typical insect: *Drosophila yakuba* Burla, 1954 (Diptera: Drosophilidae) [76]. All Hylicinae species shared seven conserved overlap regions in *trnI-trnQ* (TTG), *trnW-trnC* (AAGTCTT), *trnY-cox1* (AT), *trnK-trnD* (G), *atp8-atp6* (TGAAAATGATAA), *trnN-trnS* (A) and *trnS2-nad1* (TTAATAACTT) (Appendix A). Among these conserved overlap regions, *trnW-trnC* s (AAGTCTT) was also revealed in Ledrinae [30].

As in other published subfamilies, the AT content of the whole mtgenomes of Hylicinae indicated a strong AT bias, ranging from 75.8% in *H. paradoxa* to 77.4% in *B. fujiana* (Appendix A). Comparing the AT content of the whole mtgenome, control region, PCGs, tRNAs, and rRNAs, the rRNAs was the highest while the PCGs was the lowest for *N. tuberculatus* and *K. nativa*. For *H. paradoxa*, the AT content of the control region was the highest, and the PCGs was the lowest. In addition, the AT content of the rRNAs was the highest, and the control region was the lowest of *B. fujiana*. The composition skew analyses showed a positive AT skew and a negative GC skew in all four whole mtgenomes (Appendix A).

### 3.2. Protein-Coding Genes and Codon Usage

All 13 PCGs were detected in the newly sequenced mtgenomes and their length ranging from 10,932 bp in *H. paradoxa* to 10,962 bp in *K. nativa* (Appendix A). In all sequenced mtgenomes, nine PCGs were assigned on the J-strand, the remaining four were located on the N-strand (Appendix A). The AT skews of the PCGs ranged from −0.104 to −0.126 (Appendix A).

The majority of PCGs in the four newly sequenced mtgenomes used standard initial codons for the genetic code 5 (invertebrate mitochondrion): ATN, except for *nad5* in three of the four newly sequenced Hylicinae which started with TTG (*N. tuberculatus*, *H. paradoxa,* and *B. fujiana*, Appendix A), as seen in previous leafhoppers [29,30,61], while *nad5* in *K. nativa* started with ATT (Appendix A). After counting the terminating codon, the majority of PCGs of four mtgenomes stopped with codon TAA or TAG, whereas the remaining PCGs were stopped with a single T (Appendix A). Such incomplete termination codons also occur in other leafhoppers [29,30,61], and could accomplish completion by posttranscriptional polyadenylation [77].

The relative synonymous codon usage (RSCU) of the four Hylicinae mtgenomes were summarized in Figure 3. The four most frequently used amino acids codons were UUU (Phe), UUA (Leu2), AUA (Met), and AUU (Ile), and all of them are composed with only A and/or U.

### 3.3. Transfer and Ribosomal RNA Genes

All 22 tRNA genes of *N. tuberculatus H. paradoxa*, *B. fujiana,* and *K. nativa* mtgenomes were determined (Appendix A). The tRNAs length of these four mtgenomes was 1429 bp in *N. tuberculatus*, 1416 bp in *H. paradoxa*, 1416 bp in *B. fujiana,* and 1425 bp in *K. nativa*. The AT content of tRNA genes was slightly higher than that of the PCGs and lesser than that of the rRNAs, ranging from 75.4% to 78.9% (Appendix A). Each position of 22 tRNAs was same as relative positions in *D. yakuba* [76] and previously published cicadellid species (Table 1), except for a few leafhoppers containing tRNA rearrangements [59,60,78,79]. The sizes of these 22 tRNAs ranging from 61 (*trnW*, *trnC*, *trnD*) to 70 bp (*trnL)* in *B. fujiana*, 61 (*trnD*, *trnA*, *trnS*, *trnV*) to 70 bp (*trnQ*, *trnM*) in *H. paradoxa*, 61 (*trnA*, *trnS*, *trnV*) to 71 bp (*trnM*, *trnP*) in *K. nativa,* and from 61 (*trnG*, *trnA*, *trnE*, *trnH*) to 71 bp (*trnQ*, *trnM*) in *N. tuberculatus* (Appendix A). As shown in Appendix A, all of the 22 tRNAs had the same cloverleaf secondary structure in other Cicadellidae species [30,54,59,60,61,78]. The *trnS1* lost the DHU arm and this phenomenon probably happened very early during the evolution of the Metazoa [77]. Seven kinds of nonmatched base pairs (GU, UU, AA, AC, AG, CU, and a single C) were found in the arm structures of the tRNAs (Appendix A), while such nonmatched base pairs were also found in other published cicadellid species [29,30,43,61]. In summary, a large number of GU mismatches were the most common, which were also overrepresented as revealed in other leafhoppers [30,43,54,59,61,63].

Both of *lr*RNA (*rrnL*) and *sr*RNA (*rrnS*) genes were encoded from the N-strand in all Hylicinae mtgenomes. Among four mtgenomes of Hylicinae, the length of *rrnL* was 1180 to 1207bp, resided between *trnL1* (CUN) and *trnV*, and the *rrnS* was 730 to 740bp and located between *trnV* and the control region (Appendix A).

### 3.4. Control Region

In all mtgenomes, a large non-coding region was found (Figure 4) located between *rrnS* and *trnI* and ranges from 531 bp to 1976 bp. The A + T contents were 73.5% to 86.6% in the Hylicinae mtgenomes. Each Hylicinae had different repeat units. Three Hylicinae species had two kinds of short repeat tandem units of 11 and 131 bp in size, respectively, while *H. paradoxa* only had a tandem repeat ranging from 16 to 17 bp (Figure 4). In addition, except in *H. paradoxa*, the other three Hylicinae had Poly T/A stretches. The results showed the number of absolute tandem repeat units in control region of each Hylicinae mtgenome are different.

### 3.5. Pi values and Selection Pressures

The sliding window analysis indicated that Pi values (nucleotide diversity) were highly variable among the 13 PCGs of these Hylicinae mtgenomes (Figure 5A). The PCGs *atp8*, *nad2*, and *atp6* appear to have a comparatively high Pi values of 0.327, 0.306, and 0.261, respectively, while the PCGs *cox1*, *nad4l* and *nad1* presented relatively low Pi values of 0.185, 0.176 and 0.173 (Figure 5A). Meanwhile, the similar tendencies, that was the PCGs *atp8*, *nad2*, and *atp6* with high distances of 0.599, 0.519, and 0.406, and the PCGs *cox1*, *nad4l*, and *nad1* with low distances of 0.249, 0.234, and 0.227, severally, were produced in the analysis of genetic distance (Figure 5B).

To estimate the selection pressures of the 13 PCGs in Hylicinae species, the mean substitution rates of synonymous (Ks), non-synonymous (Ka) of the 13 PCGs among these Hylicinae mtgenomes were calculated and ranging from 0.150 to 0.984 (Figure 5B), indicating that all PCGs are under purifying selection. The Ka/Ks ratios of PCGs *atp8*, *nad2*, and *nad6* were relatively high of 0.984, 0.609, and 0.518, while the Ka/Ks ratios of *cox1*, *cox2*, and *cox3* were comparatively low of 0.150, 0.244, and 0.278, severally (Figure 5B).

Among the 13 PCGs, it was apparent that *cox1* was the slowest-evolving gene, while *atp8* was the relatively fastest-evolving gene in these Hylicinae species (Figure 5B). Moreover, the results also suggested that *nad2* and *atp8* would be capable markers for sibling hylicine species delimitation.

### 3.6. Phylogenetic Relationships

Regardless of the method used, the analyses employing different datasets (P123, P123R, and AA) resulted highly identical phylogenetic topologies and most nodes obtained strong supports (BS = 100, PP = 1) (Figure 6, Figure 7 and Figure 8). Fulgoroidea was placed at the base of Auchenorrhyncha, whereas the remaining superfamilies of Auchenorrhyncha were split into two clades: Membracoidea and Cicadoidea + Cercopoidea, which was consistent with recent studies using mtgenomes [30] and transcriptomes [80]. Our analyses corroborated the monophyly of Membracoidea [17,30,50,78,80,81]. Deltocephalinae was placed as sister group to the remainders of Membracoidea with strong support, which is also in line with previous studies [17,29,30,51,78]. For about the remaining Membracoidea, treehoppers were recovered as a monophyletic lineage and derived from Cicadellidae, which also has been supported by previous studies [17,29,50,78,81].

Recent phylogenetic studies have improved our knowledge of the subfamily-level phylogeny of paraphyletic Cicadellidae, the monophyly of nine subfamilies [30], as well as the sister-group relationship between treehoppers and Megophthalminae [29,30,78]. Our results were consistent with these findings and indicated the monophyly of expanded subfamilies (eleven) in all phylogenetic trees (Figure 6, Figure 7 and Figure 8) with strong support. Typhlocybinae was sometimes treated as sister group to all other subfamilies except Deltocephalinae [30]. Our results show the same topology in ML/BI analyses based on the datasets of P123/P123R (Figure 6 and Figure 7). However, a different relationship, Typhlocybinae + Mileewinae and Cicadellinae + (Evacanthinae + Ledrinae) formed a monophyletic group in ML/BI trees based on AA (Figure 8). The relationship, (Coelidiinae + Iassinae) + Hylicinae, was stable in our analyses (Figure 6, Figure 7 and Figure 8) and the supports were moderate to strong (BS > 86; PP = 1). The status of Eurymelinae and Mileewinae was discordant based on three datasets (Figure 6, Figure 7 and Figure 8) and the supports were low to moderate (BS = 26–92, PP < 0.92). Based on more subfamilies sampled and using different datasets, our results are somewhat inconsistent with previous molecular phylogenies [30,54,78,81]. Although we selected more comprehensive subfamily-level samples, the very short deep internal branches still appeared in our study with less than maximum support, as has been shown in other recent phylogenomic analyses of Membracoidea [17,30,81]. This phenomenon may indicate ancient rapid radiations during the Cretaceous period [17].

Within Hylicinae, the four species (*N. tuberculatus*, *H. paradoxa*, *B. fujiana*, and *K. nativa*) representing four genera of the two tribes (*H. paradoxa* representing tribe Hylicini, the remaining three representing tribe Sudrini), gathered as a monophyletic group with maximum support values. The monophyly of Hylicinae is consistent with previous studies based on morphological, anchored hybrid enrichment genomics [17]. Hylicini performs as a sister group to Sudrini in BI/ML trees based on PCGR/AA with middle to high support, while Sudrini is paraphyletic based on PCG with low support (Figure 6, Figure 7 and Figure 8). In our analyses, except for BI/ML analysis topology based on PCGs, (*H. paradoxa* + (*N. tuberculatus* + (*B. fujiana* + *K. nativa*))) was reconstructed with high posterior probabilities support values (PP > 0.98) and middle to high bootstrap support values (BS = 83–100). Unfortunately, our study also included limited taxon sampling despite having more than other related phylogenetic studies, so that we are unable to provide a robust test the monophyly of Hylicinae tribes here.

## 4. Conclusions

In our work, we report the whole mtgenome sequences of *N. tuberculatus*, *H. paradoxa*, *B. fujiana,* and *K. nativa* for Hylicinae for the first time, comparatively analyzing the mtgenomes within this subfamily and reconstructing the phylogenetic relationships of the Membracoidea. The mtgenomes of Hylicinae are highly conserved in terms of genome size, gene content and arrangement, base composition, PCG codon usage, and secondary structure of tRNAs. The variations among length of Hylicinae mtgenomes is mainly caused by the varying length of the control region. This study also shows that *nad2* and *atp8* are potential DNA markers for species delimitation for Hylicinae. Bayesian inference and maximum likelihood analyses provide the most comprehensive subfamily-rich phylogenetic analysis of Membracoidea up to now which presents a well-resolved topology and is largely congruent with previous studies. This study also demonstrates that the mtgenome provides new insight into resolving the phylogenetic relationships at family and subfamily levels. All phylogenetic analyses uniformly support Cicadellidae being paraphyletic with respect to treehoppers and all involved cicadellid subfamilies being monophyletic. The monophyly of Hylicinae is further confirmed in our study. This study also recovered a stable relationship for the subfamilies ((Coelidiinae + Iassinae) + Hylicinae) with moderate to high support. The monophyly of the two tribes, Hylicini and Sudrini, and the topology of four hylicine genera (*H. paradoxa + (N. tuberculatus + (B. fujiana + K. nativa*))) were recovered in most analyses. There are still some specific problems with Membracoidea (such as the exact placement of Eurymelinae and Mileewinae) which may be caused by the limited taxon sampling of major lineages. We therefore recommend that further research into the proposed relationships within Membracoidea should include more comprehensive taxon sampling to establish a better framework (selecting all subfamilies and tribes).

## Figures and Tables

**Figure 1 insects-11-00869-f001:**
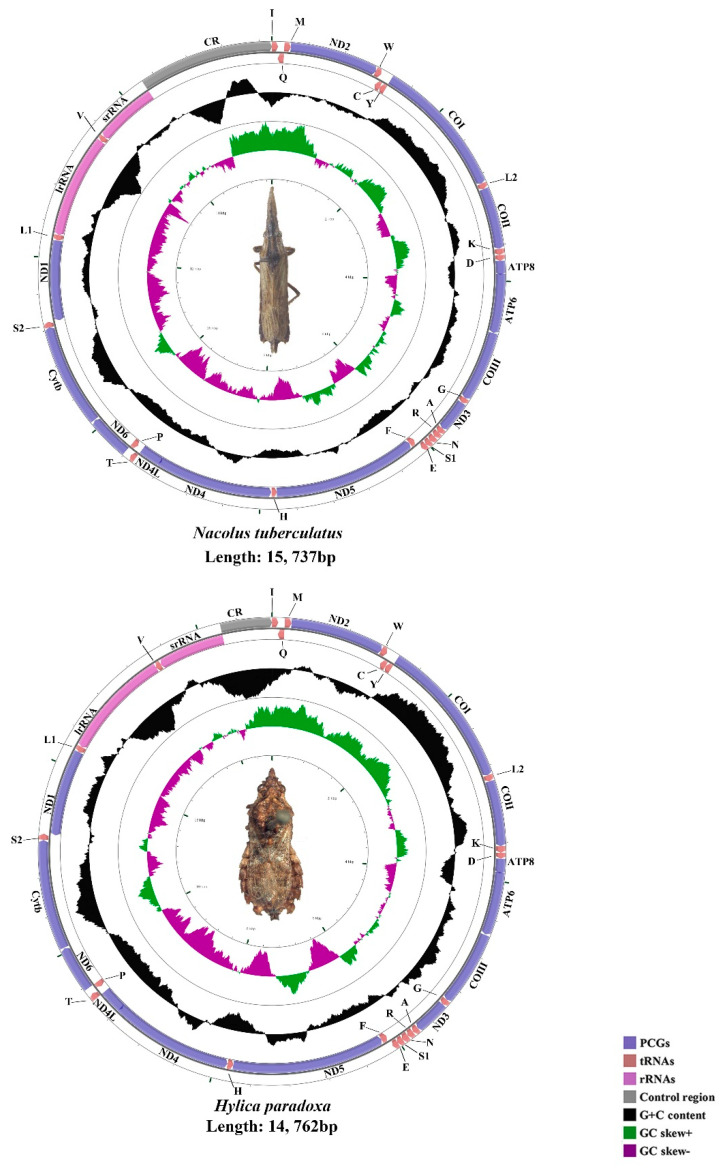
The circular 15.7 kb *N. tuberculatus* and 14.7 kb *H. paradoxa* mtgenomes.

**Figure 2 insects-11-00869-f002:**
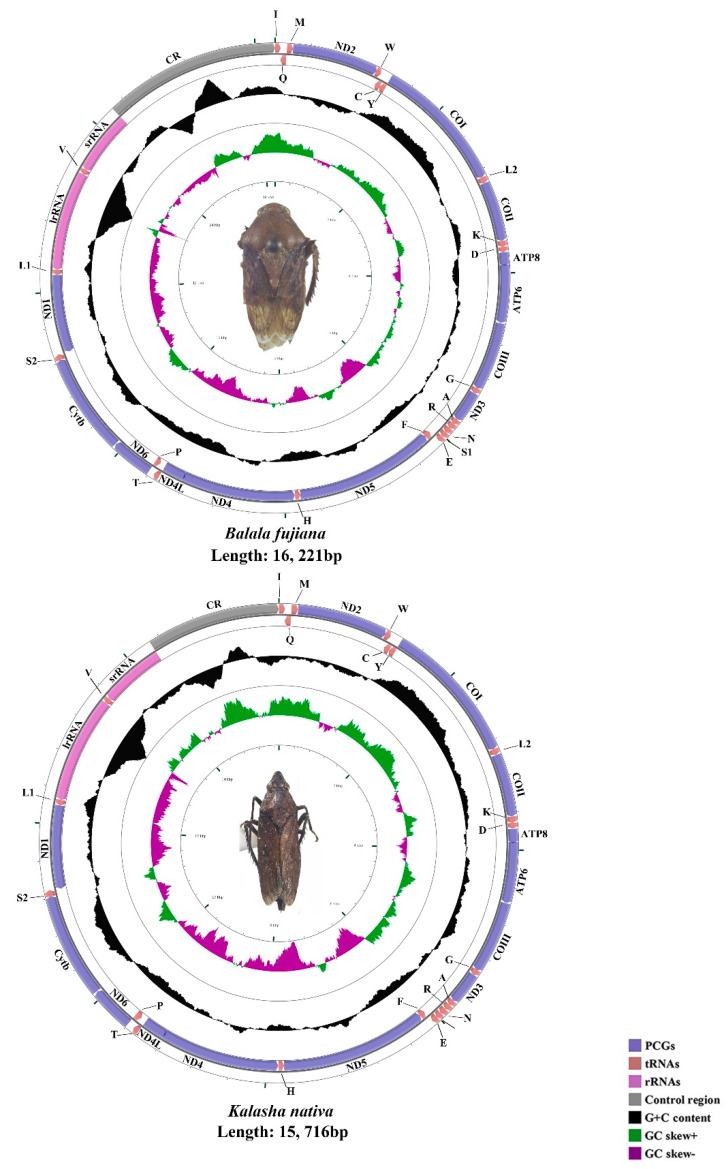
The circular 16.2 kb *B. fujiana* and 15.7 kb *K. nativa* mtgenomes.

**Figure 3 insects-11-00869-f003:**
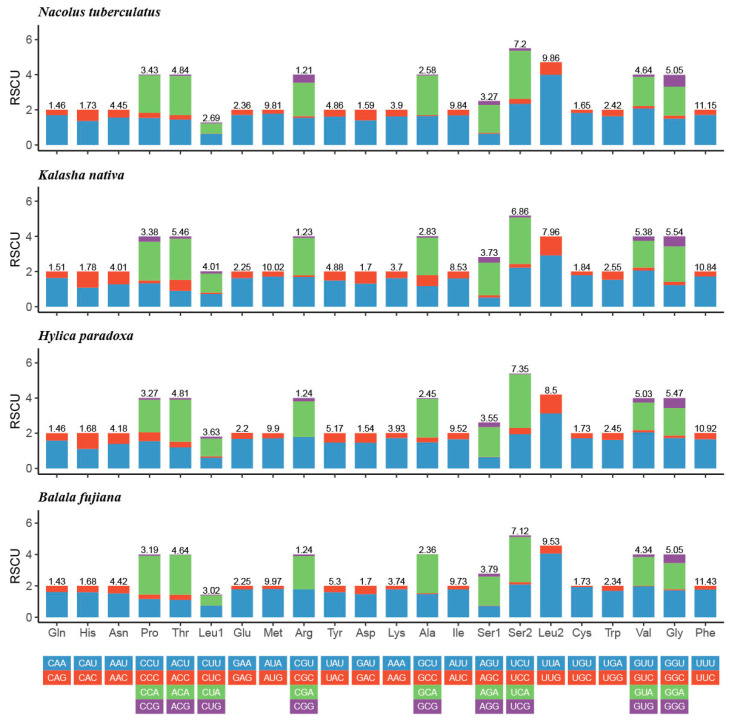
Relative synonymous codon usage (RSCU) of the mtgenomes of four Hylicinae species.

**Figure 4 insects-11-00869-f004:**
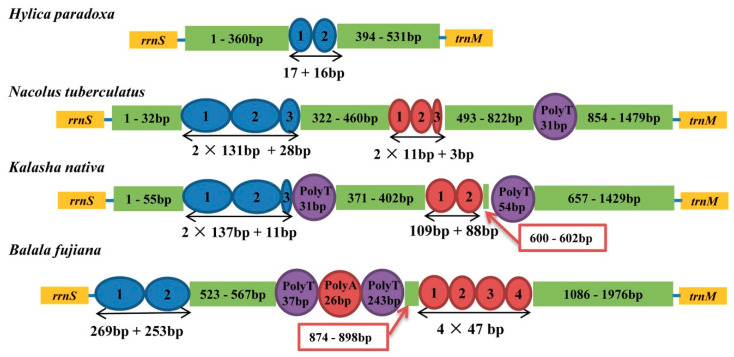
Organization of the control region in Hylicinae mtgenomes. The blue and red ovals indicate the tandem repeats; the purple and red rounds indicate the Poly T/A; the non-repeat regions are shown with green boxes.

**Figure 5 insects-11-00869-f005:**
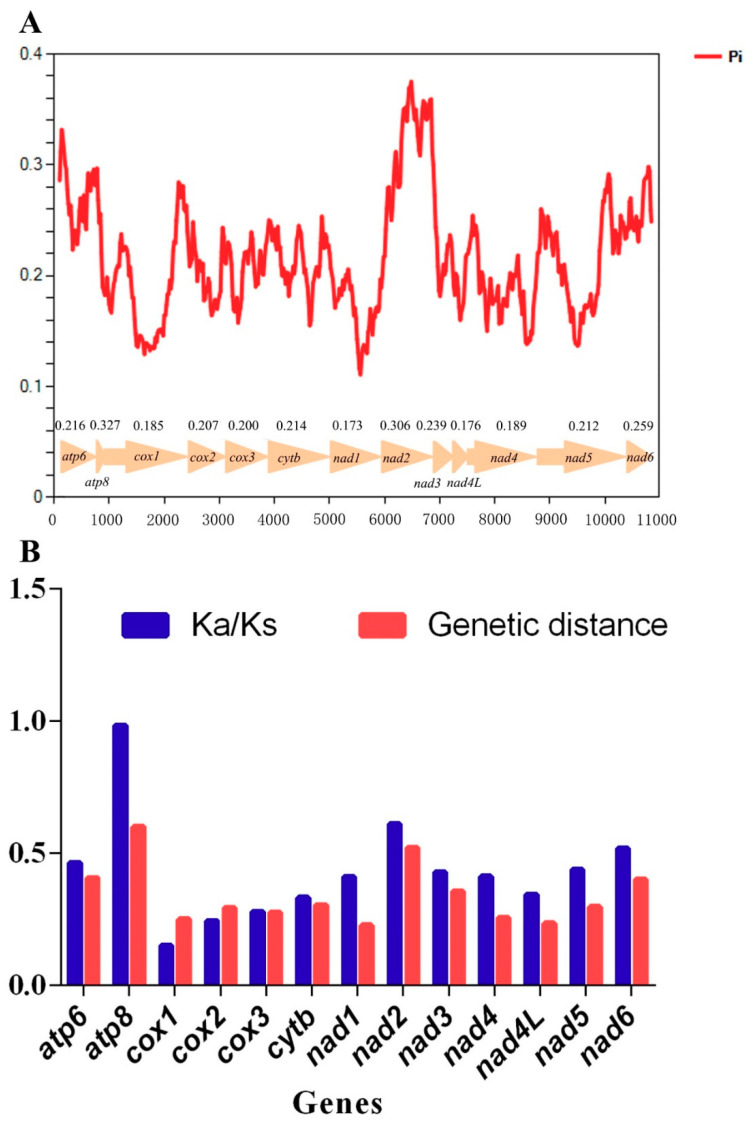
Nucleotide diversities and selection pressures of 13 protein coding genes (PCGs) in Hylicinae. (**A**) Sliding window analysis of protein-coding genes among four Hylicinae species. The red curve shows the value of Pi (nucleotide diversity). Pi value of each PCG is shown above the arrows. (**B**) Genetic distances (on average) and ratio of non-synonymous (Ka) to synonymous (Ks) substitution rates of each protein-coding gene among four Hylicinae species.

**Figure 6 insects-11-00869-f006:**
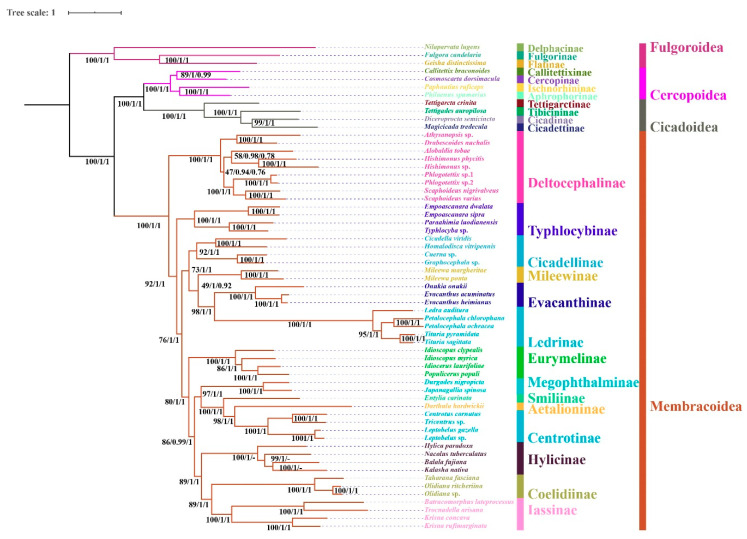
Phylogenetic tree produced by IQtree/MrBayes based on the dataset of P123R and MrBayes based on the dataset of P123. Numerals at nodes are bootstrap support values (BS) and Bayesian posterior probabilities (PP). “-” indicates the clades or species are different in Bayesian inference (BI).

**Figure 7 insects-11-00869-f007:**
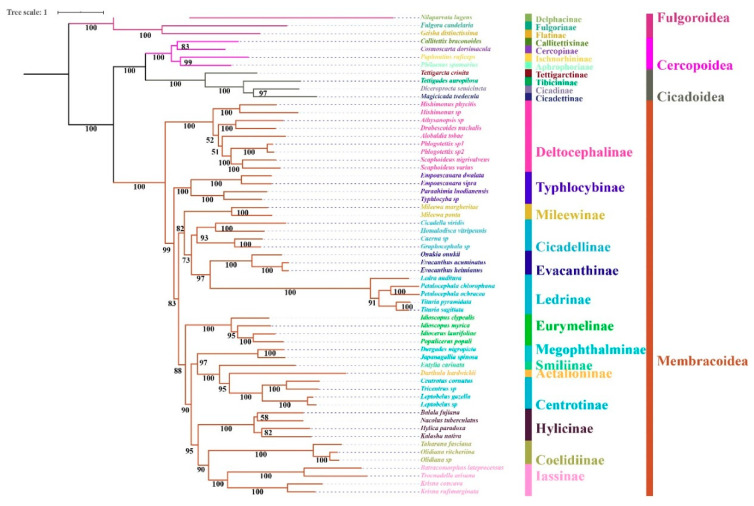
Maximum likelihood tree inferred from the dataset of PCG using IQ-TREE. Node numbers show bootstrap support values.

**Figure 8 insects-11-00869-f008:**
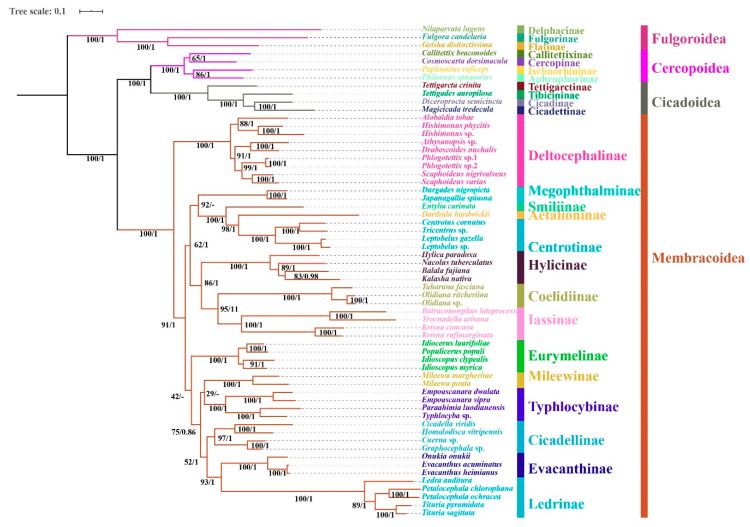
Phylogenetic tree produced by IQtree and MrBayes based on the dataset of AA. Numerals at nodes are Bayesian posterior probabilities (PP) and bootstrap support values (BS), respectively. “-” indicates the clades or species are different.

**Table 1 insects-11-00869-t001:** Information of samples for phylogenetic analyses.

Superfamily	Family/Subfamily	Species	Accession Number	Reference
**Outgroup**				
Fulgoroidea	Fulgoridae/Fulgorinae	*Fulgora candelaria*	NC_019576	[26]
	Flatidae/Flatinae	*Geisha distinctissima*	NC_012617	[42]
	Delphacidae/Delphacinae	*Nilaparvata lugens*	NC_021748	[43]
Cercopoidea	Cercopidae/Cercopinae	*Cosmoscarta dorsimacula*	NC_040115	Unpublished
	Cercopidae/Callitettixinae	*Callitettix braconoides*	NC_025497	[44]
	Cercopidae/Ischnorhininae	*Paphnutius ruficeps*	NC_021100	[45]
	Aphrophoridae/Aphrophorinae	*Philaenus spumarius*	NC_005944	[46]
Cicadoidea	Cicadidae/Cicadinae	*Diceroprocta semicincta*	KM000131	Unpublished
	Cicadidae/Cicadettinae	*Magicicada tredecula*	MH937705	[24]
	Cicadidae/Tibicininae	*Tettigades auropilosa*	KM000129	Unpublished
	Tettigarctidae/Tettigarctinae	*Tettigarcta crinita*	MG737758	[47]
**Ingroup**				
Membracoidea	Aetalionidae/Aetalioninae	*Darthula hardwickii*	NC_026699	[48]
	Membracidae/Centrotinae	*Leptobelus gazella*	NC_023219	[49]
	Membracidae/Centrotinae	*Centrotus cornutus*	KX437728	[50]
	Membracidae/Centrotinae	*Tricentrus* sp.	KY039115	[51]
	Membracidae/Centrotinae	*Leptobelus* sp.	JQ910984	[25]
	Membracidae/Smiliinae	*Entylia carinata*	NC_033539	[52]
	Cicadellidae/Cicadellinae	*Cuerna* sp.	KX437741	[50]
	Cicadellidae/Cicadellinae	*Graphocephala* sp.	KX437740	[50]
	Cicadellidae/Cicadellinae	*Cicadella viridis*	KY752061	Unpublished
	Cicadellidae/Cicadellinae	*Homalodisca vitripennis*	NC_006899	Unpublished
	Cicadellidae/Coelidiinae	*Taharana fasciana*	NC_036015	[53]
	Cicadellidae/Coelidiinae	*Olidiana* sp.	KY039119	Unpublished
	Cicadellidae/Coelidiinae	*Olidiana_ritcheriina*	MK738125	[54]
	Cicadellidae/Deltocephalinae	*Hishimonus phycitis*	KX437727	[50]
	Cicadellidae/Deltocephalinae	*Hishimonus* sp.	KX437735	[50]
	Cicadellidae/Deltocephalinae	*Phlogotettix* sp. 1	KY039135	[51]
	Cicadellidae/Deltocephalinae	*Phlogotettix* sp. 2	KX437721	[50]
	Cicadellidae/Deltocephalinae	*Scaphoideus varius*	KY817245	[51]
	Cicadellidae/Deltocephalinae	*Scaphoideus nigrivalveus*	KY817244	[51]
	Cicadellidae/Deltocephalinae	*Alobaldia tobae*	KY039116	[51]
	Cicadellidae/Deltocephalinae	*Athysanopsis* sp.	KX437726	[50]
	Cicadellidae/Deltocephalinae	*Drabescoides nuchalis*	NC_028154	[55]
	Cicadellidae/Evacanthinae	*Evacanthus heimianus*	MG813486	[56]
	Cicadellidae/Evacanthinae	*Evacanthus acuminatus*	MK948205	[57]
	Cicadellidae/Evacanthinae	*Onukia onukii*	MK251119	[58]
	Cicadellidae/Eurymelinae	*Populicerus populi*	NC_039427	[59]
	Cicadellidae/Eurymelinae	*Idiocerus laurifoliae*	NC_039741	[59]
	Cicadellidae/Eurymelinae	*Idioscopus clypealis*	NC_0396f42	[60]
	Cicadellidae/Eurymelinae	*Idioscopus myrica*	MH492317	[59]
	Cicadellidae/Hylicinae	*Kalasha nativa*	MW218662	This study
	Cicadellidae/Hylicinae	*Nacolus tuberculatus*	MW218663	This study
	Cicadellidae/Hylicinae	*Balala fujiana*	MW218661	This study
	Cicadellidae/Hylicinae	*Hylica paradoxa*	MW218660	This study
	Cicadellidae/Iassinae	*Trocnadella arisana*	NC036480	[61]
	Cicadellidae/Iassinae	*Krisna rufimarginata*	NC046068	[61]
	Cicadellidae/Iassinae	*Krisna concava*	MN577635	[61]
	Cicadellidae/Iassinae	*Batracomorphus lateprocessus*	MG813489	[61]
	Cicadellidae/Ledrinae	*Tituria pyramidata*	MN920440	[62]
	Cicadellidae/Ledrinae	*Ledra auditura*	MK387845	[63]
	Cicadellidae/Ledrinae	*Petalocephala ochracea*	KX437734	[50]
	Cicadellidae/Ledrinae	*Petalocephala chlorophana*	MT610899	[30]
	Cicadellidae/Ledrinae	*Tituria sagittata*	MT610900	[30]
	Cicadellidae/Mileewinae	*Mileewa margheritae*	MT483998	[64]
	Cicadellidae/Mileewinae	*Mileewa ponta*	MT497465	[65]
	Cicadellidae/Megophthalminae	*Japanagallia spinosa*	NC_035685	[66]
	Cicadellidae/Megophthalminae	*Durgades nigropicta*	NC_035684	[66]
	Cicadellidae/Typhlocybinae	*Typhlocyba* sp.	KY039138	[51]
	Cicadellidae/Typhlocybinae	*Empoascanara dwalata*	MT350235	Unpublished
	Cicadellidae/Typhlocybinae	*Empoascanara sipra*	MN604278	[67]
	Cicadellidae/Typhlocybinae	*Paraahimia luodianensis*	NC047464	[68]

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
