# Peer review of "The Complete Mitochondrial Genome of Four Hylicinae (Hemiptera: Cicadellidae): Structural Features and Phylogenetic Implications"

_insects, 2020, doi:10.3390/insects11120869_

Round 1

Reviewer 1 Report

The manuscript "The complete mitochondrial genome of four Hylicinae (Hemiptera: Cicadellidae): structural features and phylogenetic implications" presents important scientific information that deserves to be published in insects. Before publication, I suggest minor modifications:

Page 2, line 55

Change “Wolfella” for “Wolfella Spinola, 1850”

Change “Karasekia” for “Karasekia Melichar, 1926”

Page 2, line 88

Change “(N. tuberculatus, H. paradoxa, B. fujiana and K. nativa) for “[Nacolus tuberculatus (Walker, 1858), Hylica paradoxa Stål, 1863, Balala fujiana Tang & Zhang, 2020 and Kalasha native Distant1908]

Page 3, line 96

Change “N. tuberculatus was captured...” for “Nacolus tuberculatus was captured…”

Page 3, line 97

Change “H. paradoxa was collected…” for “Hylica paradoxa was collected…”

Page 3, line 108

Change “Evacanthus heimianus” for “Evacanthus heimianus Kuoh, 1981”

Page 5, line 166

Change “Drosophila yakuba” for “Drosophila yakuba Burla, 1954 (Diptera: Drosophilidae)”

Fig. 7 and 8: I cannot read the names of the species in the phylogeny. Please enlarge the image while maintaining focus.

Author Response

’insects-1015904-coverletter-Author's Reply1‘ is attached.

Reviewer 2 Report

Dr Tang and Collaborators have obtained the complete sequence of the nuclear genome of four hyalicine hemipterans. They provide a description of the genomic features of the new species and a phylogenetic analysis of these in the context of several representatives of Membracoidea whose sequences are available on GenBank.

The structure of the study is fairly standard in the field, but seems to have been conducted following reasonably high standards. The results per se do not allow for a conclusive statement on the phylogenetic relationships among the species studied, but provide a reasonably supported hypothesis for the position of the subfamily.

I think that, following some clarification and some correction to the figure and supplementary material, the manuscript could be a suitable contribution for Insects.

Line 63: please reword. Monophily cannot be assessed based on one single representative.

Line 96: please give coordinates. As databases expand and species are renamed, this may become an essential information.

Section 2.2: Please provide details of the sequencing and assembly. Were sequences of the four species tagged, separately sequenced, or mixed? If mixed, how was each sequence assigned to a species? Can the authors claim these are high quality assemblies (unfortunately low quality assemblies are flooding the databases and the utmost care is needed here)?

Line 130: how were these species chosen among the 150 available? Was the choice based on representativity, quality, .... ? Why were not all included?

Table 1: accessions for new sequences appear (to me) in the wrong column.

Line140: what does 'corrected artificially' mean. Please explain. Modifying the alignment manually after g-blocks can introduce major errors if a base is moved across two neighbouring blocks.

Line 146: which blocks were proposed to PartitionFinder to be joined in the search of optimal partitions?

Line 151: 5 to 20 million generations may not be awfully high for a dataset of this size. Can the Authors claim they are sufficient based on the observation of trace files?

Line 161: did the Authors compare the length and boundaries of PCGs of the new species with homologous sequences from known genomes based on some statistics or at least a visual inspection of alignments? This is a key step to guard from errors in the assembly that may reflect in truncated/incomplete genes or frameshift errors.

Figure 1-2: what does each circle represent?

Line 176: from the discussion below I argue that the AT rich region was compared as well.

Line 241: was this analysis conducted on entire (aligned) PCGs or on the phylogenetic dataset (i.e. after excluding hypervariable regions using g-blocks)? If a comparison is to be made in terms of gene variability, the entire PCGs need to be used, otherwise a major (and variable) bias may be observed towards conservation (i.e. g-blocks excluded hypervariable regions selectively).

Line 248: please reword. This is not about estimating 'rates of evolution'.

Line 255-256: Why?

Figures 6-7-8: Please remove the color legend, as this information is already provided in full on the right.

Figures 6-7-8: Taxon names are too small to read. Please modify.

There are some formatting errors in the supplementary (these may be due to wrong conversion among software). Please check. S7: font; S5: no columns headers. I suggest submitting these as .pdf to ensure cross platform compatibility.

The list of software and kits in S8 is uncommon. I suggest giving credits in the main text using citations, where needed.

There seems to be some confusion between 'unpublished material', that actually include all figures and a copy of the supplementary, and the supplementary material themselves.

Author Response

’insects-1015904-coverletter-Author's Reply2‘ is attached.
